# Adversarial Adaptive Sampling: Unify PINN and Optimal Transport for the Approximation of PDEs

**Kejun Tang,**[*] **Jiayu Zhai,**[*] **Xiaoliang Wan**[◇]**, Chao Yang**[§]
PKU-Changsha Institute for Computing and Digital Economy
Institute of Mathematical Sciences, ShanghaiTech University
[◇]Department of Mathematics and Center for Computation & Technology, Louisiana State University
[§]School of Mathematical Sciences, Peking University & PKU-Changsha Institute for Computing and Digital Economy
`tangkejun@icode.pku.edu.cn`, `zhaijy@shanghaitech.edu.cn`
`xlwan@math.lsu.edu`, `chao_yang@pku.edu.cn`

## Abstract

Solving partial differential equations (PDEs) is a central task in scientific computing. Recently, neural network approximation of PDEs has received increasing attention due to its flexible meshless discretization and its potential for high-dimensional problems. One fundamental numerical difficulty is that random samples in the training set introduce statistical errors into the discretization of the loss functional which may become the dominant error in the final approximation, and therefore overshadow the modeling capability of the neural network. In this work, we propose a new minmax formulation to optimize simultaneously the approximate solution, given by a neural network model, and the random samples in the training set, provided by a deep generative model. The key idea is to use a deep generative model to adjust the random samples in the training set such that the residual induced by the neural network model can maintain a smooth profile in the training process. Such an idea is achieved by implicitly embedding the Wasserstein distance between the residual-induced distribution and the uniform distribution into the loss, which is then minimized together with the residual. A nearly uniform residual profile means that its variance is small for any normalized weight function such that the Monte Carlo approximation error of the loss functional is reduced significantly for a certain sample size. The adversarial adaptive sampling (AAS) approach proposed in this work is the first attempt to formulate two essential components, minimizing the residual and seeking the optimal training set, into one minmax objective functional for the neural network approximation of PDEs.

## 1 Introduction

Partial differential equations (PDEs) are widely used to model physical phenomena. Typically, obtaining analytical solutions to PDEs is intractable, and thus numerical methods (e.g., finite element methods (Elman et al., 2014)) have to be developed to approximate the solutions of PDEs. However, classical numerical methods can be computationally infeasible for high-dimensional PDEs due to the curse of dimensionality or computationally expensive for parametric low-dimensional PDEs (Xiu & Karniadakis, 2003; Ghosh et al., 2022; Yin et al., 2023; Zhai et al., 2022). To alleviate these difficulties, machine learning (ML) techniques, e.g., physics-informed neural networks (PINN) (Raissi et al., 2019) and deep Ritz method (E & Yu, 2018), have been adapted to approximate PDEs as surrogate models and have received increasing attention (Han et al., 2018; Zhu & Zabaras, 2018; Zhu et al., 2019; Weinan, 2021; Karniadakis et al., 2021). The basic idea of deep learning methods for approximating PDEs is to encode the information of PDEs in neural networks through a proper loss functional, which will be discretized by collocation points in the computational domain and subsequently minimized to determine an optimal model parameter (Raissi et al., 2019; E & Yu, 2018; Sirignano & Spiliopoulos, 2018; Zhu et al., 2019).

---

[*]Co-first Author

The collocation points are crucial to effectively train neural networks for PDEs because they provide an approximation of the loss functional. In the community of computer vision or natural language processing, it is well known that the performance of ML models is highly dependent on the quality of data (i.e., the training set). Similarly, if the selected collocation points fail to yield an accurate approximation of the loss functional, it is not surprising that the trained neural network will suffer a large generalization error, especially when the solution has low regularity or the problem dimension is large. As shown in (Tang et al., 2023; Wu et al., 2023), if the collocation points in the training set are refined according to a proper error indicator, the accuracy can be dramatically improved. This is similar to classical adaptive methods such as the adaptive finite element method (Morin et al., 2002; Mekchay & Nochetto, 2005). In this work, we propose a new framework, called *adversarial adaptive sampling* (AAS), that simultaneously optimizes the loss functional and the training set to seek neural network approximation for PDEs through a minmax formulation. More specifically, we minimize the residual and meanwhile push the residual-induced distribution to a uniform distribution. To do this, we introduce a deep generative model into the AAS formulation, which not only provides random samples for the discretization of the loss functional, but also plays the role of the critic in WGAN (Arjovsky et al., 2017; Gulrajani et al., 2017). In the maximization step, the deep generative model helps identify the difference in a Wasserstein distance between the residual-induced distribution and a uniform distribution; in the minimization step, such a difference is minimized together with the residual. This way, variance reduction is achieved once the residual profile is smoothed and the loss functional can be better approximated by a fixed number of random samples, which yields a more effective optimal model parameter, i.e., a more accurate neural network approximation of the PDE solution.

The main contributions of this paper are as follows.

- We unify PINN and optimal transport into an adversarial adaptive sampling framework, which provides a new perspective on neural network methods for solving PDEs.
- We develop a theoretical understanding of AAS and propose a simple but effective algorithm.

## 2 PINN AND ITS STATISTICAL ERRORS

The PDE problem considered here is: find $u \in \mathscr{F} : \Omega \mapsto \mathbb{R}$ where $\mathscr{F}$ is a proper function space defined on a computational domain $\Omega \in \mathbb{R}^D$, such that

$$
\begin{aligned}
\mathcal{L}u(\boldsymbol{x}) &= s(\boldsymbol{x}), \quad \forall \boldsymbol{x} \in \Omega \\
\flat u(\boldsymbol{x}) &= g(\boldsymbol{x}), \quad \forall \boldsymbol{x} \in \partial\Omega,
\end{aligned}
\tag{1}
$$

where $\mathcal{L}$ is a partial differential operator (e.g., the Laplace operator $\Delta$), $\flat$ is a boundary operator (e.g., the Dirichlet boundary), $s$ is the source function, and $g$ represents the boundary conditions. In the framework of PINN (Raissi et al., 2019), the solution $u$ of equation 1 is approximated by a neural network $u_{\boldsymbol{\theta}}(\boldsymbol{x})$ (parameterized with $\boldsymbol{\theta}$). The parameters $\boldsymbol{\theta}$ is determined by minimizing the following loss functional

$$
\begin{aligned}
J\left(u_{\boldsymbol{\theta}}\right) &= J_r(u_{\boldsymbol{\theta}}) + \gamma J_b(u_{\boldsymbol{\theta}}) \quad \text{with} \\
J_r(u_{\boldsymbol{\theta}}) &= \int_{\Omega} |r(\boldsymbol{x};\boldsymbol{\theta})|^2 d\boldsymbol{x} \ \text{ and } \ J_b(u_{\boldsymbol{\theta}}) = \int_{\partial\Omega} |b(\boldsymbol{x};\boldsymbol{\theta})|^2 d\boldsymbol{x},
\end{aligned}
\tag{2}
$$

where $r(\boldsymbol{x};\boldsymbol{\theta}) = \mathcal{L}u_{\boldsymbol{\theta}}(\boldsymbol{x}) - s(\boldsymbol{x})$, and $b(\boldsymbol{x};\boldsymbol{\theta}) = \flat u_{\boldsymbol{\theta}}(\boldsymbol{x}) - g(\boldsymbol{x})$ are the residuals that measure how well $u_{\boldsymbol{\theta}}$ satisfies the partial differential equations and the boundary conditions, respectively, and $\gamma > 0$ is a penalty parameter. To optimize this loss functional with respect to $\boldsymbol{\theta}$, we need to discretize the integral defined in equation 2 numerically. Let $\mathsf{S}_{\Omega} = \{\boldsymbol{x}_{\Omega}^{(i)}\}_{i=1}^{N_r}$ and $\mathsf{S}_{\partial\Omega} = \{\boldsymbol{x}_{\partial\Omega}^{(i)}\}_{i=1}^{N_b}$ be two sets of uniformly distributed collocation points on $\Omega$ and $\partial\Omega$ respectively. We then minimize the following empirical loss in practice

$$
J_N\left(u_{\boldsymbol{\theta}}\right) = \frac{1}{N_r} \sum_{i=1}^{N_r} r^2(\boldsymbol{x}_{\Omega}^{(i)};\boldsymbol{\theta}) + \gamma \frac{1}{N_b} \sum_{i=1}^{N_b} b^2(\boldsymbol{x}_{\partial\Omega}^{(i)};\boldsymbol{\theta}),
\tag{3}
$$

which can be regarded as the Monte Carlo (MC) approximation of $J(u_{\boldsymbol{\theta}})$ subject to a statistical error of $O(N^{-1/2})$ with $N$ being the sample size. Let $u_{\boldsymbol{\theta}_N^*}$ be the minimizer of the empirical loss $J_N(u_{\boldsymbol{\theta}})$

and $u_{\boldsymbol{\theta}^*}$ be the minimizer of the original loss functional $J(u_{\boldsymbol{\theta}})$. We can decompose the error into two parts as follows

$$\mathbb{E}\left(\left\|u_{\boldsymbol{\theta}_N^*} - u\right\|_\Omega\right) \leq \mathbb{E}\left(\left\|u_{\boldsymbol{\theta}_N^*} - u_{\boldsymbol{\theta}^*}\right\|_\Omega\right) + \left\|u_{\boldsymbol{\theta}^*} - u\right\|_\Omega,$$

where $\mathbb{E}$ denotes the expectation operator and the norm $\|\cdot\|_\Omega$ corresponds to the function space $\mathscr{F}$ for $u$. One can see that the total error of neural network approximation for PDEs comes from two main aspects: the approximation error and the statistical error. The approximation error is dependent on the model capability of neural networks, while the statistical error originates from the collocation points. Uniformly distributed collocation points are not effective for training neural networks if the solution has low regularity (Tang et al., 2022; 2023; Wu et al., 2023) since the effective sample size of the MC approximation of $J(u_{\boldsymbol{\theta}})$ is significantly reduced by the large variance induced by the low regularity. For high-dimensional problems, information becomes more sparse or localized due to the curse of dimensionality, which shares some similarities with the low-dimensional problems of low regularity. Adaptive sampling is needed. In this work, we propose a new framework to optimize both the approximation solution and the training set.

## 3 Adversarial Adaptive Sampling

Adversarial adaptive sampling (AAS) includes two components to be optimized. One is a neural network $u_{\boldsymbol{\theta}}$ for approximating the PDE solution, and another is a probability density function (PDF) model $p_{\boldsymbol{\alpha}}$ (parameterized with $\boldsymbol{\alpha}$) for sampling. Unlike the deep adaptive sampling method (DAS) presented in (Tang et al., 2023), in AAS, we simultaneously optimize the two models through an adversarial training procedure, which provides a new perspective to understand the role of random samples for the neural network approximation of PDEs.

### 3.1 A minmax formulation

The adversarial adaptive sampling approach can be formulated as the following minmax problem

$$\min_{\boldsymbol{\theta}} \max_{p_{\boldsymbol{\alpha}} \in V} \mathcal{J}(u_{\boldsymbol{\theta}}, p_{\boldsymbol{\alpha}}) = \int_\Omega r^2(\boldsymbol{x}; \boldsymbol{\theta}) p_{\boldsymbol{\alpha}}(\boldsymbol{x}) d\boldsymbol{x} + \gamma J_b(u_{\boldsymbol{\theta}}), \tag{4}$$

where $V$ is a function space that defines a proper constraint on $p_{\boldsymbol{\alpha}}(\boldsymbol{x})$. The choice of $V$ will be specified in sections 3.2 and 3.3 in terms of the theoretical understanding and numerical implementation of AAS.

The main difference between $\mathcal{J}(u_{\boldsymbol{\theta}}, p_{\boldsymbol{\alpha}})$ and $J(u_{\boldsymbol{\theta}})$ in equation 2 is that the weight function for the integration of $r^2(\boldsymbol{x}; \boldsymbol{\theta})$ is relaxed to $p_{\boldsymbol{\alpha}}(\boldsymbol{x})$ from a uniform one. First of all, such a relaxation can also be applied to $J_b(\cdot)$. In this work, we focus on the integration of $r^2$ for simplicity and assume that $J_b(\cdot)$ is well approximated by a prescribed set $\mathsf{S}_{\partial\Omega}$. Indeed, some penalty-free techniques (Berg & Nyström, 2018; Sheng & Yang, 2021) can be employed to remove $J_b(\cdot)$. Second, $p_{\boldsymbol{\alpha}}(\boldsymbol{x}) > 0$ is regarded as a PDF on $\Omega$, and an extra constraint on $p_{\boldsymbol{\alpha}}$ is necessary. Otherwise, the maximization step will simply yield a delta measure, i.e.,

$$\delta(\boldsymbol{x} - \boldsymbol{x}_0) = \arg\max_{p > 0, \int_\Omega p d\boldsymbol{x} = 1} \int_\Omega r^2(\boldsymbol{x}; \boldsymbol{\theta}) p(\boldsymbol{x}) d\boldsymbol{x},$$

where $\boldsymbol{x}_0 = \arg\max_{\boldsymbol{x} \in \Omega} r^2(\boldsymbol{x}; \boldsymbol{\theta})$. Nevertheless, the region of large residuals is of particular importance for adaptive sampling. Third, the maximization in terms of $p_{\boldsymbol{\alpha}}$ is important numerically rather than theoretically. Indeed, if the statistical error does not exist and the model $u_{\boldsymbol{\theta}}$ includes the exact PDE solution, the minimum of $r^2$ is always reached at 0 as long as $p_{\boldsymbol{\alpha}}$ is positive on $\Omega$. To reduce the statistical error induced by the random samples from $p_{\boldsymbol{\alpha}}$, we expect a small variance $\mathrm{Var}(r^2)$ in terms of $p_{\boldsymbol{\alpha}}$. If the variance of the Monte Carlo integration for $r^2$ is smaller than the variance of $r^2$ in terms of the uniform distribution, the accuracy of the Monte Carlo approximation will be improved for a fixed sample size, which yields a more accurate solution of PDEs (Tang et al., 2023). To obtain a small $\mathrm{Var}(r^2)$, the profile of the residual needs to be nearly uniform. So an effective training strategy should not only minimize the residual but also endeavor to maintain a smooth profile of the residual, in other words, the two models $u_{\boldsymbol{\theta}}$ and $p_{\boldsymbol{\alpha}}$ need to work together. See Figure 1 for an informal description of the approach.

We will model $p_{\boldsymbol{\alpha}}$ using a bounded KRnet, which defines an invertible mapping $f_{\boldsymbol{\alpha}}(\cdot) : I^D \to I^D$ with $I = [-1, 1]$ and yields a normalizing flow model. In this work, we consider $\Omega = I^D$ for simplicity. The bounded KRnet can be achieved by adding a logistic transformation layer (Tang et al., 2023) or a new coupling layer proposed in (Zeng et al., 2023). Let $\boldsymbol{z} = f_{\boldsymbol{\alpha}}(\boldsymbol{x})$ and $p_{\boldsymbol{Z}}(\boldsymbol{z})$ be a prior PDF. We define $p_{\boldsymbol{\alpha}}$ as

Figure 1: Two neural network models are simultaneously trained in the adversarial adaptive sampling framework. The residual is minimized and finally becomes "uniform", while the collocation points are updated and finally become nonuniform.

$$p_{\boldsymbol{\alpha}}(\boldsymbol{x}) = p_{\boldsymbol{Z}}(f_{\boldsymbol{\alpha}}(\boldsymbol{x}))|\nabla_{\boldsymbol{x}} f_{\boldsymbol{\alpha}}|.$$

Depending on a priori knowledge of the problem, the prior $p_{\boldsymbol{Z}}(\boldsymbol{z})$ can be chosen as a uniform distribution or more general models such as Gaussian mixture model.

## 3.2 UNDERSTANDING OF AAS

For simplicity and clarity, we remove $J_b(u_{\boldsymbol{\theta}})$ and consider

$$\min_{\boldsymbol{\theta}} \max_{p \in V} \mathcal{J}(u_{\boldsymbol{\theta}}, p) = \int_{\Omega} r^2(\boldsymbol{x}; \boldsymbol{\theta}) p(\boldsymbol{x}) d\boldsymbol{x}. \tag{5}$$

We choose $V$ as

$$V := \{p(\boldsymbol{x}) | \|p\|_{\mathrm{Lip}} \le 1, \, 0 \le p(\boldsymbol{x}) \le M\},$$

where $M$ is a positive number. We define a bounded metric

$$d_M(\boldsymbol{x}, \boldsymbol{y}) = \min\{M, d(\boldsymbol{x}, \boldsymbol{y})\}, \quad \boldsymbol{x}, \boldsymbol{y} \in \mathbb{R}^D,$$

where $d(\boldsymbol{x}, \boldsymbol{y}) = \|\boldsymbol{x} - \boldsymbol{y}\|_2$ is the Euclidean metric in $\mathbb{R}^D$. Without loss of generality, let $\Omega$ be a compact subset of $\mathbb{R}^D$ with total Lebesgue measure 1, and $\mu$ and $\nu$ two probability measures on $\Omega$. The *Wasserstein distance* $d_{W^M}(\mu, \nu)$ between $\mu$ and $\nu$ for the metric $d_M(\boldsymbol{x}, \boldsymbol{y})$ is

$$d_{W^M}(\mu, \nu) = \inf_{\pi \in \Pi(\Omega \times \Omega)} \int_{\Omega \times \Omega} d_M(\boldsymbol{x}, \boldsymbol{y}) \, d\pi(\boldsymbol{x}, \boldsymbol{y}),$$

where $\Pi(\Omega \times \Omega)$ is the collection of all joint probability measures on $\Omega \times \Omega$. The dual form (see e.g. (Villani, 2003), Theorem 1.14 and Remark 1.15 on Page 34) of $d_{W^M}$ is

$$d_{W^M}(\mu, \nu) = \sup \left\{ \left. \int_{\Omega} \phi(\boldsymbol{x}) \, d(\mu - \nu)(\boldsymbol{x}) \right| 0 \le \phi(\boldsymbol{x}) \le \|d_M\|_{\infty} = M, \text{ and } \|\phi\|_{\mathrm{Lip}} \le 1 \right\}, \tag{6}$$

where $\|\phi\|_{\mathrm{Lip}}$ is the Lipschitz norm of function $\phi$. We now reformulate the maximization problem as

$$\sup_{p \in V} \int_{\Omega} r^2(\boldsymbol{x}; \theta) p(\boldsymbol{x}) d\boldsymbol{x}$$

$$= \sup_{p \in V} \int_{\Omega} r^2(\boldsymbol{x}; \theta) p(\boldsymbol{x}) d\boldsymbol{x} - \int_{\Omega} r^2(\boldsymbol{x}; \theta) d\boldsymbol{x} \int_{\Omega} p(\boldsymbol{x}) d\boldsymbol{x} + \int_{\Omega} r^2(\boldsymbol{x}; \theta) d\boldsymbol{x} \int_{\Omega} p(\boldsymbol{x}) d\boldsymbol{x}$$

$$\le \int_{\Omega} r^2(\boldsymbol{x}; \theta) d\boldsymbol{x} \left( \sup_{p \in V} \left[ \int_{\Omega} p(\boldsymbol{x}) d\mu_r - \int_{\Omega} p(\boldsymbol{x}) d\mu_u \right] + \sup_{p \in V} \int_{\Omega} p(\boldsymbol{x}) d\boldsymbol{x} \right)$$

$$\le (d_{W^M}(\mu_r, \mu_u) + M) \int_{\Omega} r^2(\boldsymbol{x}; \boldsymbol{\theta}) \, d\boldsymbol{x},$$

where $\mu_r$ and $\mu_u$ indicate the probability measures on $\Omega$ induced by $r^2(\boldsymbol{x})$ and the uniform distribution on $\Omega$ respectively. It can be shown that the constant $M$ exists if we modify the function space $V$ as

$$\hat{V} = \{p(\boldsymbol{x}) | \|p\|_{\mathrm{Lip}} \le 1, \, p(\boldsymbol{x}) \ge 0, \int_{\Omega} p(\boldsymbol{x}) d\boldsymbol{x} = 1\},$$

where $p(\boldsymbol{x})$ can then be regarded as a PDF. It is seen that the upper bound includes both the loss of the standard PINN and the Wasserstein distance between the residual induced distribution $\mu_r$ and the uniform distribution $\mu_u$. For any $u$, the existence of function $\tilde{u}$, which has the same total residual loss as $u$ and a uniform residual profile, is theoretically guaranteed (for detailed construction and

these properties of $\tilde{u}$, please see the proof of Theorem 4 in Appendix A.2). This ensures that we can simultaneously reduce the residual and the Wasserstein distance between the (renormalized) residual and the uniform distribution. Once the residual profile is smoothed, variance reduction is achieved such that the Monte Carlo approximation of $\mathcal{J}(u_{\boldsymbol{\theta}}, p)$ will be more accurate for a fixed sample size. This eventually reduces the statistical error of the approximate PDE solution.

We now summarize our main analytical results. Consider

$$\inf_{u} \sup_{p \in \hat{V}} \mathcal{J}(u, p) = \int_{\Omega} r^2(u(\boldsymbol{x})) p(\boldsymbol{x}) \, d\boldsymbol{x}, \tag{7}$$

with the following assumption.

**Assumption A1.** The operator $r$ in equation 7 is a surjection from a function space $E_1(\mathbb{R}^D)$ to $C_c^{\infty}(\Omega)$, the class of $C^{\infty}$ functions that are compactly supported on $\Omega$.

In general, $E_1(\mathbb{R}^D)$ can be any function space, such as space of neural networks, smooth functions, or Sobolev spaces. And this assumption means for any smooth function $f \in C_c^{\infty}(\Omega)$, equation $r^2(u^*) = f$ admits some solution $u^*$. For example, if $r$ is Laplacian $\Delta$, the assumption means we can find a solution for $\Delta u = f$ for any $f$ in $C_c^{\infty}(\Omega)$. With this assumption, we can prove the following main theorem for the min-max problem equation 7 (for the detailed proof, please see the Section A.2 in the supplementary material),

**Theorem 1.** *Let $\mu$ be the Lebesgue measure on $\mathbb{R}^D$, which represents the uniform probability distribution on $\Omega$. In addition, we assume Assumption A1 holds. Then the optimal value of the min-max problem equation 7 is 0. Moreover, there is a sequence $\{u_n\}_{n=1}^{\infty}$ of functions with $r(u_n) \neq 0$ for all $n$, such that it is an optimization sequence of equation 7, namely,*

$$\lim_{n \to \infty} \mathcal{J}(u_n, p_n) = 0,$$

*for some sequence of functions $\{p_n\}_{n=1}^{\infty}$ satisfying the constraints in equation 7. Meanwhile, this optimization sequence has the following two properties:*

1. *The residual sequence $\{r(u_n)\}_{n=1}^{\infty}$ of $\{u_n\}_{n=1}^{\infty}$ converges to 0 in $L^2(d\mu)$.*

2. *The renormalized squared residual distributions*

$$d\nu_n \triangleq \frac{r^2(u_n)}{\int_{\Omega} r^2(u_n(\boldsymbol{x})) \, d\boldsymbol{x}} \, d\mu(\boldsymbol{x})$$

   *converge to the uniform distribution $\mu$ in the Wasserstein distance $d_{WM}$.*

### 3.3 IMPLEMENTATION OF AAS

In the previous section, we have shown that $p_{\boldsymbol{\alpha}}(\boldsymbol{x})$ and $u_{\boldsymbol{\theta}}(\boldsymbol{x})$ in equation 4 play a similar role as the critic and generator in WGAN (Arjovsky et al., 2017; Gulrajani et al., 2017). The generator of WGAN minimizes the Wasserstein distance between two distributions; PINN minimizes the residual; AAS achieves a tradeoff between the minimization of the residual and the minimization of the Wasserstein distance between the residual-induced distribution and the uniform distribution. From the implementation point of view, a particular difficulty is the constraint $\|p\|_{\mathrm{Lip}} \leq 1$ induced by the function space $\hat{V}$. In this work, we propose a weaker constraint that can be easily implemented. We consider

$$\min_{\boldsymbol{\theta}} \max_{\substack{p_{\boldsymbol{\alpha}} > 0, \\ \int_{\Omega} p_{\boldsymbol{\alpha}}(\boldsymbol{x}) d\boldsymbol{x} = 1}} \mathcal{J}(u_{\boldsymbol{\theta}}, p_{\boldsymbol{\alpha}}) = \int_{\Omega} r^2(\boldsymbol{x}; \boldsymbol{\theta}) p_{\boldsymbol{\alpha}}(\boldsymbol{x}) d\boldsymbol{x} - \beta \int_{\Omega} |\nabla_{\boldsymbol{x}} p_{\boldsymbol{\alpha}}(\boldsymbol{x})|^2 d\boldsymbol{x}, \tag{8}$$

where we use a $H_1$ regularization term to replace explicit control on the Lipschitz condition. The constraints on a PDF are naturally satisfied because $p_{\boldsymbol{\alpha}}$ is a normalizing flow model. $p_{\boldsymbol{\alpha}}(\boldsymbol{x}) > 0$ as long as the prior is positive since $f_{\boldsymbol{\alpha}}(\cdot)$ is an invertible mapping. It can be shown that the maximizer for a fixed $u_{\boldsymbol{\theta}}$ is uniquely determined by the following elliptic equation

$$\begin{cases} 2\beta \nabla^2 p^* + r^2(\boldsymbol{x}; \boldsymbol{\theta}) - \frac{1}{|\Omega|} \int_{\Omega} r^2(\boldsymbol{x}; \boldsymbol{\theta}) d\boldsymbol{x} = 0, & \boldsymbol{x} \in \Omega, \\ \frac{\partial p^*}{\partial \boldsymbol{n}} = 0, & \boldsymbol{x} \in \partial\Omega. \end{cases} \tag{9}$$

In the deep learning framework, the neural networks are in general (particularly when solving PDEs) differentiable. So the regularity constraint $\|p^*\|_{\text{Lip}} \leq M$ is equivalent to $\|\nabla p^*\|_\infty \leq M$ on a compact set $\Omega$. Thus, we can adjust the penalty parameter $\beta$ to implicitly control this regularity. Such a choice is demonstrated to be empirically sufficient since we focus on PDE approximation instead of PDF approximation.

To update $\boldsymbol{\theta}$ at the minimization step, we approximate the first term of $\mathcal{J}(u_{\boldsymbol{\theta}}, p_{\boldsymbol{\alpha}})$ in equation 8 using Monte Carlo methods:

$$\int_\Omega r^2 \left[u_{\boldsymbol{\theta}}(\boldsymbol{x})\right] p_{\boldsymbol{\alpha}}(\boldsymbol{x}) d\boldsymbol{x} \approx \frac{1}{m} \sum_{i=1}^m r^2 \left[u_{\boldsymbol{\theta}}(\boldsymbol{x}_{\boldsymbol{\alpha}}^{(i)})\right], \tag{10}$$

where $\boldsymbol{x}_{\boldsymbol{\alpha}}^{(i)}$ can be generated from the probability density $p_{\boldsymbol{\alpha}}$ efficiently thanks to the invertible mapping $f_{\boldsymbol{\alpha}}(\cdot)$. To update $\boldsymbol{\alpha}$ at the maximization step, we approximate $\mathcal{J}(u_{\boldsymbol{\theta}}, p_{\boldsymbol{\alpha}})$ by importance sampling:

$$\mathcal{J}(u_{\boldsymbol{\theta}}, p_{\boldsymbol{\alpha}}) \approx \frac{1}{m} \sum_{i=1}^m \frac{r^2 \left[u_{\boldsymbol{\theta}}(\boldsymbol{x}_{\boldsymbol{\alpha}'}^{(i)})\right] p_{\boldsymbol{\alpha}}(\boldsymbol{x}_{\boldsymbol{\alpha}'}^{(i)})}{p_{\boldsymbol{\alpha}'}(\boldsymbol{x}_{\boldsymbol{\alpha}'}^{(i)})} - \beta \cdot \frac{1}{m} \sum_{i=1}^m \frac{|\nabla_{\boldsymbol{x}} p_{\boldsymbol{\alpha}}(\boldsymbol{x}_{\boldsymbol{\alpha}'}^{(i)})|^2}{p_{\boldsymbol{\alpha}'}(\boldsymbol{x}_{\boldsymbol{\alpha}'}^{(i)})}, \tag{11}$$

where $p_{\boldsymbol{\alpha}'}$ is a PDF model with known parameters $\boldsymbol{\alpha}'$ and each $x_{\boldsymbol{\alpha}'}^{(i)}$ is a sample drawn from $p_{\boldsymbol{\alpha}'}$. Using equation 10 and equation 11, we can compute the gradient with respect to $\boldsymbol{\theta}$ and $\boldsymbol{\alpha}$, and the parameters can be updated by gradient-based optimization methods (e.g., Adam (Kingma & Ba, 2017)). The training procedure is similar to GAN (Goodfellow et al., 2014) and can be summarized in Algorithm 1, where we let $p_{\boldsymbol{\alpha}'} = p_{\boldsymbol{\alpha}_k}$ in equation 11, i.e., the PDF model from the last step is used for importance sampling when computing $\mathcal{J}(u_{\boldsymbol{\theta}}, p_{\boldsymbol{\alpha}})$.

---

**Algorithm 1** AAS for PDEs

---

**Input:** Initial $p_{\boldsymbol{\alpha}}$ and $u_{\boldsymbol{\theta}}$, maximal iteration $M$, batch size $m$, initial training set $\mathsf{S}_{\Omega,0} = \{\boldsymbol{x}_{\boldsymbol{\alpha}_0}^{(i)}\}_{i=1}^{N_r}$
    and $\mathsf{S}_{\partial\Omega,0} = \{\boldsymbol{x}_{\partial\Omega,0}^{(i)}\}_{i=1}^{N_b}$.
1: **for** $k = 0, \ldots, M$ **do**
2:     **for** $j$ steps **do**
3:         Sample $m$ samples from $\mathsf{S}_{\Omega,k}$ and sample $m$ samples from $\mathsf{S}_{\partial\Omega,k}$.
4:         Update $u_{\boldsymbol{\theta}}$ by descending the stochastic gradient of $\mathcal{J}(\boldsymbol{\theta}, \boldsymbol{\alpha})$ (see equation 10).
5:     **end for**
6:     **for** $j$ steps **do**
7:         Sample $m$ samples from $\mathsf{S}_{\Omega,k}$.
8:         Update $p_{\boldsymbol{\alpha}}$ by ascending the stochastic gradient of $\mathcal{J}(\boldsymbol{\theta}, \boldsymbol{\alpha})$ (see equation 11).
9:     **end for**
10:    Generate $\mathsf{S}_{\Omega,k+1} \subset \Omega$ through $p_{\boldsymbol{\alpha}_k}$.
11: **end for**
**Output:** $u_{\boldsymbol{\theta}}$

---

## 4 RELATED WORK

There is a lot of related work, and we summarize the most related lines of this work.

**Adaptive sampling**. Adaptive sampling methods have been receiving increasing attention in solving PDEs with deep learning methods. The basic idea of such methods is to define a proper error indicator (Wu et al., 2023; Yu et al., 2022) for refining collocation points in the training set, in which sampling approaches (Gao & Wang, 2023) (e.g., Markov Chain Monte Carlo) or deep generative models (Tang et al., 2023) are often invoked. To this end, an additional deep generative model (e.g., normalizing flow models), or classical PDF model (e.g., Gaussian mixture models (Gao et al., 2022; Jiao et al., 2023)) for sampling is usually required, which is similar to this work. However, there are some crucial differences between existing approaches and the proposed adversarial adaptive sampling (AAS) framework. First, in AAS, the evolution of the residual-induced distribution has a clear path. That is, this residual-induced distribution is pushed to a uniform distribution during

training. Because minimizing the Wasserstein distance between the residual-induced distribution and the uniform distribution is naturally embedded in the loss functional in the proposed adversarial sampling framework. Second, unlike the existing methods, our AAS method admits an adversarial training style like in WGAN, which is the first time to minimize the residual and seek the optimal training set simultaneously for PINN.

**Adversarial training**. In (Zang et al., 2020), the authors proposed a weak formulation with primal and adversarial networks, where the PDE problem is converted to an operator norm minimization problem derived from the weak formulation. Although the adversarial training procedure is employed in (Zang et al., 2020), it does not involve the training set but the function space. Introducing one or more discriminator networks to construct adversarial training is studied in (Zeng et al., 2022), where the discriminator is used for the reward that PINN predicts mistakes. However, this approach does not optimize the training set but implicitly assigns higher weights for samples with large pointwise residuals through adversarial training.

# 5 NUMERICAL RESULTS

We use some benchmark test problems presented in (Tang et al., 2023) to demonstrate the proposed method. All models are set to be the same as those in DAS-PINNs (Tang et al., 2023) and trained by the Adam method (Kingma & Ba, 2017). The hyperparameters of neural networks are set to be the same as those in DAS-PINNs (Tang et al., 2023). For comparison, we also implement the DAS algorithm (Tang et al., 2023) and the RAR algorithm (Lu et al., 2021; Yu et al., 2022) as the baseline models. The training of neural networks is performed on a Geforce RTX 3090 GPU with TensorFlow 2.0. The codes of all examples will be released on GitHub once the paper is accepted.

## 5.1 ONE-PEAK PROBLEM

We start with the following equation which is a benchmark test problem for adaptive finite element methods (Mitchell, 2013; Morin et al., 2002):

$$-\Delta u(\boldsymbol{x}) = s(\boldsymbol{x}) \quad \text{in } \Omega,$$
$$u(\boldsymbol{x}) = g(\boldsymbol{x}) \quad \text{on } \partial\Omega, \tag{12}$$

where $\boldsymbol{x} = [x_1, x_2]^\mathsf{T}$ and the computation domain is $\Omega = [-1, 1]^2$. The following reference solution is given by

$$u(x_1, x_2) = \exp\left(-1000[(x_1 - 0.5)^2 + (x_2 - 0.5)^2]\right),$$

which has a peak at $(0.5, 0.5)$ and decreases rapidly away from $(0.5, 0.5)$. The reference solution is imposed on the boundary. The source term $s(\boldsymbol{x})$ is derived by the exact solution and is listed in the supplementary A.5. A uniform meshgrid with size $256 \times 256$ in $[-1, 1]^2$ is generated and the error is defined to be the mean square error on these grid points. From Figure 2(a), it can be seen that our AAS method can give an accurate approximation for this peak test problem. Note that the uniform sampling strategy is not suitable for this test problem as studied in (Tang et al., 2023). The training behaviour for different regularization parameters (i.e., $\beta$) is shown in Figure 2(a) and Figure 2(b). It can be seen that the error behavior is similar for $\beta = 5, 10, 20$. Figure Figure 2(c) shows the evolution of the residual variance and the training set during training for $\beta = 5$, where the variance decreases as the training step increases and the training set finally concentrates on $(0.5, 0.5)$ with a heavy tail. The comparison of different adaptive sampling methods is presented in Table 1, which also included the results of the following test problems.

## 5.2 TWO-PEAK PROBLEM

We next consider the following equation

$$-\nabla \cdot [u(\boldsymbol{x})\nabla v(\boldsymbol{x})] + \nabla^2 u(\boldsymbol{x}) = s(\boldsymbol{x}) \quad \text{in } \Omega,$$
$$u(\boldsymbol{x}) = g(\boldsymbol{x}) \quad \text{on } \partial\Omega, \tag{13}$$

where $\boldsymbol{x} = [x_1, x_2]^\mathsf{T}$, $v(\boldsymbol{x}) = x_1^2 + x_2^2$, and the computation domain is $\Omega = [-1, 1]^2$. Following (Tang et al., 2023), the exact solution of equation 13 is set to be as

$$u(x_1, x_2) = \mathrm{e}^{-1000[(x_1-0.5)^2+(x_2-0.5)^2]} + \mathrm{e}^{-1000[(x_1+0.5)^2+(x_2+0.5)^2]},$$

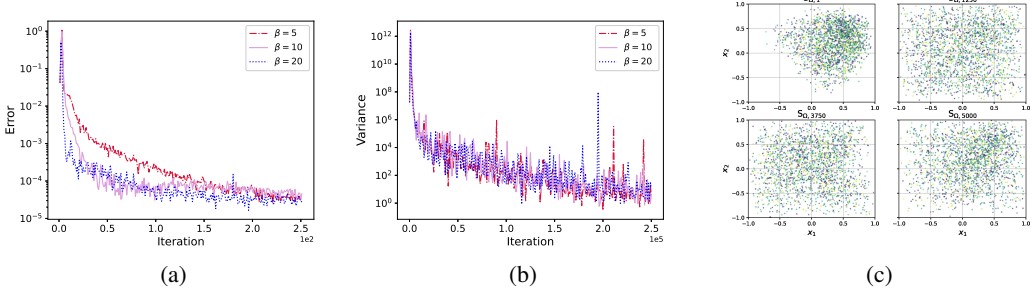

(a)          (b)          (c)

Figure 2: The results for the peak test problem. (a) The error behaviour. (b) The variance behavior. (c) The evolution of the training set.

which has two peaks at the points $(0.5, 0.5)$ and $(-0.5, -0.5)$. Here, the Dirichlet boundary condition on $\partial\Omega$ is given by the exact solution. From Figure 3(a) and Figure 3(b), it can be seen that our AAS method can give an accurate approximation for this two-peak test problem. The error behavior for different regularization parameters (i.e., $\beta$) is shown in Figure 3(c). Figure 4 shows the evolution of the residual variance and the training set during training for $\beta = 5$, where the residual variance decreases as the training step increases and the training set finally concentrates on $(-0.5, -0.5)$ and $(0.5, 0.5)$ with a heavy tail.

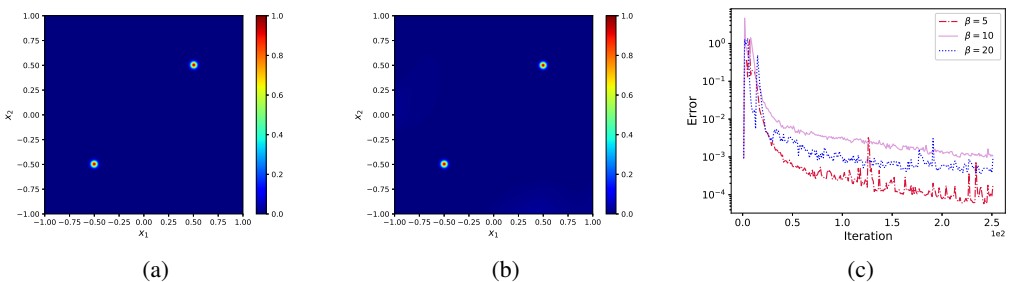

(a)          (b)          (c)

Figure 3: The results for the two-peak test problem. (a) The exact solution. (b) AAS approximation. (c) The error behavior.

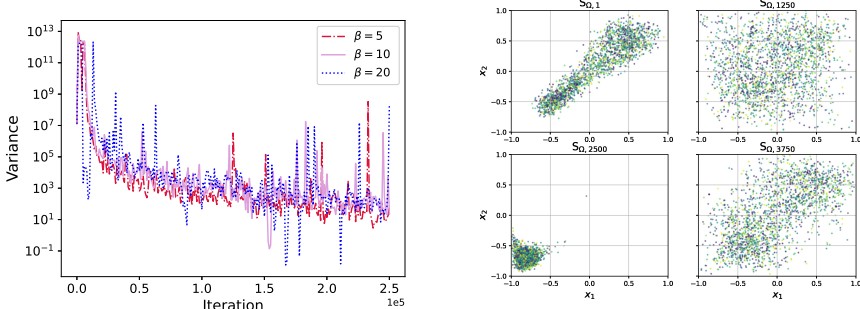

Figure 4: The evolution of the residual variance and the training set for the two-peak test problem. Left: The variance behavior. Right: The evolution of the training set.

### 5.3 HIGH-DIMENSIONAL NONLINEAR EQUATION

In this part, we consider the following ten-dimensional nonlinear partial differential equation

$$-\Delta u(\boldsymbol{x}) + u(\boldsymbol{x}) - u^3(\boldsymbol{x}) = s(\boldsymbol{x}), \quad \boldsymbol{x} \text{ in } \Omega = [-1, 1]^{10}$$
$$u(\boldsymbol{x}) = g(\boldsymbol{x}), \quad \boldsymbol{x} \text{ on } \partial\Omega. \tag{14}$$

The exact solution is $u(\boldsymbol{x}) = \mathrm{e}^{-10\|\boldsymbol{x}\|_2^2}$ and the Dirichlet boundary condition on $\partial\Omega$ is imposed by the exact solution. The source term $s(\boldsymbol{x})$ is derived by the exact solution and is listed in the supplementary A.5. The error is defined to be the same as in (Tang et al., 2023). Figure 5 shows the results of the ten-dimensional nonlinear test problem. Specifically, Figure 5(a) shows the error behavior during training for different regularization parameters, and Figure 5(b) shows the evolution of the residual variance. Figure 5(c) shows the samples during the adversarial training process, where we select the components $x_1$ and $x_2$ for visualization. We have also checked the other components, and the results are similar. It is seen that the training set finally becomes nonuniform to get a small residual variance. The results of different adaptive sampling strategies for the three test problems are summarized in Table 1.

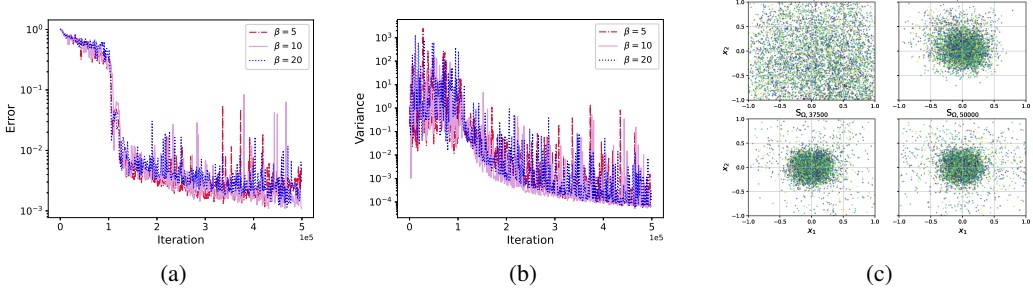

| (a) | (b) | (c) |

Figure 5: The results of the ten-dimensional nonlinear test problem. (a) The error behavior. (b) The variance behaviour. (c) The evolution of the training set, $x_1 - x_2$ plane ($\beta = 10$).

Table 1: Error comparison of adaptive sampling methods

| Method | Test problem / PDE equation 12 | PDE equation 13 | PDE equation 14 |
|---|---|---|---|
| PINN | 9.74e-04 | 3.22e-02 | 1.01 |
| RAR (Lu et al., 2021) | - | - | 9.83e-01 |
| DAS-G (Tang et al., 2023) | 3.75e-04 | 1.51e-03 | 9.55e-03 |
| DAS-R (Tang et al., 2023) | 1.93e-04 | 6.21e-03 | 1.26e-02 |
| AAS (this work) | **2.97e-05** | **1.09e-04** | **1.31e-03** |

## 6 CONCLUSIONS

We developed a novel adversarial adaptive sampling (AAS) approach that unifies PINN and optimal transport for neural network approximation of PDEs. With AAS, the evolution of the training set can be investigated in terms of the optimal transport theory, and numerical results have demonstrated the importance of random samples for training PINN more effectively.

**Acknowledgments:** K. Tang has been supported by the China Postdoctoral Science Foundation grant 2022M711730. J. Zhai is supported by the start-up fund of ShanghaiTech University (2022F0303-000-11). X. Wan has been supported by NSF grant DMS-1913163. C. Yang has been supported by NSFC grant 12131002.

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

## A    APPENDIX

We add the proof of Theorem 1 and additional numerical experiments here.

### A.1    PRELIMINARIES FROM OPTIMAL TRANSPORT THEORY

**Definition 2.** Suppose $X$ is a metric space equipped with the metric $d(\boldsymbol{x}, \boldsymbol{y})$, and $\mu$ and $\nu$ are two probability measures on $X$. The *Wasserstein distance* (as known as the *Kantorovich–Rubinstein metric*) $d_{W^d}(\mu, \nu)$ between to probability measures $\mu$ and $\nu$ for the metric function $d(\boldsymbol{x}, \boldsymbol{y})$ is defined to be

$$d_{W^d}(\mu, \nu) = \inf_{\pi \in \Pi(X \times X)} \int_{X \times X} d(\boldsymbol{x}, \boldsymbol{y}) \, d\pi(\boldsymbol{x}, \boldsymbol{y}),$$

where $\Pi(X \times X)$ is the collection of all probability measure on $X \times X$ such that

$$\pi(A \times X) = \mu(A), \quad \pi(X \times B) = \nu(B)$$

for all measurable sets $A, B \subset X$.

For the analysis of the adaptive algorithm in this work, we consider the metric $d_M(\boldsymbol{x}, \boldsymbol{y})$ induced by the Euclidean metric $d(\boldsymbol{x}, \boldsymbol{y}) = \|\boldsymbol{x} - \boldsymbol{y}\|_2$

$$d_M(\boldsymbol{x}, \boldsymbol{y}) = \min\{M, d(\boldsymbol{x}, \boldsymbol{y})\}, \quad \boldsymbol{x}, \boldsymbol{y} \in X.$$

Then the metric $d_M(\boldsymbol{x}, \boldsymbol{y})$ is always bounded by $M$ (reachable, namely $\|d_M\|_\infty = M$). We denote the Wasserstein distance for $d_M(\boldsymbol{x}, \boldsymbol{y})$ by $d_{W^M}(\cdot, \cdot)$.

According to the optimal transport theory, the Wasserstein distance can be described by its dual form (see e.g. (Villani, 2003), Theorem 1.14 and Remark 1.15 on Page 34).

**Theorem 3** (Kantorovich-Rubinstein theorem). *Let $X$ be a Polish space and let $d$ be a lower semi-continuous metric on $X$. Let $\| \cdot \|_{Lip}$ denote the Lipschitz norm of a function defined as*

$$\|\phi\|_{Lip} = \sup_{\boldsymbol{x} \neq \boldsymbol{y}} \frac{|\phi(\boldsymbol{x}) - \phi(\boldsymbol{y})|}{d(\boldsymbol{x}, \boldsymbol{y})}.$$

*Then*

$$d_{W^M}(\mu, \nu) = \sup \left\{ \int_X \phi(\boldsymbol{x}) \, d(\mu - \nu)(\boldsymbol{x}) \, \middle| \, 0 \leq \phi(\boldsymbol{x}) \leq \|d_M\|_\infty = M, \text{ and } \|\phi\|_{Lip} \leq 1 \right\}.$$

In this work, we restrict ourselves on a compact domain $X = \Omega \subset \mathbb{R}^D$ of learning, and without loss of generality, we assume the Lebesgue measure of $\Omega$ is $1$.

### A.2    THE FIRST CONVERGENCE THEOREM AND ITS PROOF

**Theorem 4.** *Let $\mu$ be the Lebesgue measure on $X$, which represents the uniform probability distribution on $\Omega$. In addition, we assume Assumption A1 holds.*

*Then the optimal value of the min-max problem equation 5 is $0$. Moreover, there is a sequence $\{u_n\}_{n=1}^\infty$ of functions with $r(u_n) \neq 0$ for all $n$, such that it is an optimization sequence of problem equation 5, namely,*

$$\lim_{n \to \infty} \mathcal{J}(u_n, p_n) = 0. \tag{15}$$

*for some sequence of functions $\{p_n\}_{n=1}^\infty \subset V$. Meanwhile, this optimization sequence has the following two properties:*

  1. *The residual sequence $\{r(u_n)\}_{n=1}^\infty$ of $\{u_n\}_{n=1}^\infty$ converges to $0$ in $L^2(d\mu)$.*

  2. *The renormalized squared residual distributions*

$$d\nu_n \triangleq \frac{r^2(u_n)}{\int_\Omega r^2(u_n(\boldsymbol{x})) \, d\boldsymbol{x}} \, d\mu \tag{16}$$

*converge to the uniform distribution $\mu$ in the Wasserstein distance $d_{W^M}$.*

*Proof.* Consider a minimizing sequence $u_n, n = 1, 2, \ldots$ of

$$\inf_u \int_\Omega r^2(u(\boldsymbol{x})) \, d\boldsymbol{x}, \tag{17}$$

where without loss of generality, we can assume that $\int_\Omega r^2(u_n(\boldsymbol{x})) \, d\boldsymbol{x} \leq \frac{1}{n}$.

Now

$$\sup_{\substack{\|p\|_{\mathrm{Lip}} \leq 1 \\ 0 \leq p \leq M}} \mathcal{J}(u_n, p)$$

$$= \sup_{\substack{\|p\|_{\mathrm{Lip}} \leq 1 \\ 0 \leq p \leq M}} \Big[ \int_\Omega r^2(u_n(\boldsymbol{x}))p(\boldsymbol{x}) \, d\boldsymbol{x} - \int_\Omega r^2(u_n(\boldsymbol{x})) \, d\boldsymbol{x} \int_\Omega p(\boldsymbol{x}) \, d\boldsymbol{x} + \int_\Omega r^2(u_n(\boldsymbol{x})) \, d\boldsymbol{x} \int_\Omega p(\boldsymbol{x}) \, d\boldsymbol{x} \Big]$$

$$\leq \int_\Omega r^2(u_n(\boldsymbol{x})) \, d\boldsymbol{x} \Big( \sup_{\substack{\|p\|_{\mathrm{Lip}} \leq 1 \\ 0 \leq p \leq M}} \Big[ \int_\Omega p(\boldsymbol{x}) \, d\nu_n(\boldsymbol{x}) - \int_\Omega p(\boldsymbol{x}) \, d\boldsymbol{x} \Big] + \sup_{\substack{\|p\|_{\mathrm{Lip}} \leq 1 \\ 0 \leq p \leq M}} \int_\Omega p(\boldsymbol{x}) \, d\boldsymbol{x} \Big)$$

$$= (d_{W^M}(\nu_n, \mu) + M) \int_\Omega r^2(u_n(\boldsymbol{x})) \, d\boldsymbol{x}. \tag{18}$$

By the assumption of the theorem, for each $n$, we can find a function $\tilde{u}_n(\boldsymbol{x})$ so that the Wasserstein distance $d_{W^M}(\tilde{\nu}_n, \mu) \leq \frac{1}{n}$, where $\tilde{\nu}_n$ is the measure defined as in equation 16 by replacing $u_n(\boldsymbol{x})$ with $\tilde{u}_n(\boldsymbol{x})$. In fact, for each $n$, we can find, by partition of unity, a sequence of functions in $C_c^\infty(\Omega)$ converging to $\mathbb{1}_\Omega$ in the Sobolev norm of $W^{k,1}$ (See for example (Evans, 2010)). So we can find a function $w_n$ in $C_c^\infty(\Omega)$, such that $\|w_n(\boldsymbol{x}) - \mathbb{1}_\Omega(\boldsymbol{x})\|_1 \leq \frac{1}{n}$ on $\Omega$. Since $r$ is a surjection, there is some $\tilde{u}_n(\boldsymbol{x})$ so that

$$r^2(\tilde{u}_n) = w_n \int_\Omega r^2(u_n(\boldsymbol{x})) \, d\boldsymbol{x},$$

and

$$\int_\Omega r^2(\tilde{u}_n) \, d\boldsymbol{x} = \int_\Omega w_n(\boldsymbol{x}) \, d\boldsymbol{x} \int_\Omega r^2(u_n(\boldsymbol{x})) \, d\boldsymbol{x}$$

$$\leq \Big(1 + \int_\Omega \mathbb{1}_\Omega(\boldsymbol{x}) \, d\boldsymbol{x}\Big) \int_\Omega r^2(u_n(\boldsymbol{x})) \, d\boldsymbol{x}$$

$$= 2 \int_\Omega r^2(u_n(\boldsymbol{x})) \, d\boldsymbol{x}.$$

This means $\{\tilde{u}_n\}_{n=1}^\infty$ is also a minimizing sequence of equation 17, and it yields

$$d_{W^M}(\tilde{\nu}_n, \mu) = \sup_{\substack{\|p\|_{\mathrm{Lip}} \leq 1 \\ 0 \leq p \leq M}} \Big[ \int_\Omega p(\boldsymbol{x}) \, d\tilde{\nu}_n(\boldsymbol{x}) - \int_\Omega p(\boldsymbol{x}) \, d\boldsymbol{x} \Big]$$

$$= \sup_{\substack{\|p\|_{\mathrm{Lip}} \leq 1 \\ 0 \leq p \leq M}} \int_\Omega p(\boldsymbol{x}) \Big[ \frac{r^2(\tilde{u}_n)(\boldsymbol{x})}{\int_\Omega r^2(\tilde{u}_n(\boldsymbol{x})) \, d\boldsymbol{x}} - \mathbb{1}_\Omega(\boldsymbol{x}) \Big] \, d\boldsymbol{x}$$

$$= \sup_{\substack{\|p\|_{\mathrm{Lip}} \leq 1 \\ 0 \leq p \leq M}} \int_\Omega p(\boldsymbol{x}) \big[ w_n(\boldsymbol{x}) - \mathbb{1}_\Omega(\boldsymbol{x}) \big] \, d\boldsymbol{x}$$

$$\leq \frac{M}{n}.$$

So we get from equation 18 that

$$0 \leq \lim_{n \to \infty} \sup_{\substack{\|p\|_{\mathrm{Lip}} \leq 1 \\ 0 \leq p \leq M}} \mathcal{J}(\tilde{u}_n, p) \leq \lim_{n \to \infty} 4M \int_\Omega r^2(u_n) \, d\boldsymbol{x} = 0,$$

which means that $\{\tilde{u}_n\}_{n=1}^\infty$ is also a minimizing sequence of equation 5, that is,

$$\lim_{n \to \infty} \mathcal{J}(\tilde{u}_n, p_n) = 0., \tag{15}$$

for some sequence of functions $\{p_n\}_{n=1}^\infty \subset V$. Meanwhile, we have the following properties of $\tilde{u}_n$:

1. The residual sequence $\{r(\tilde{u}_n)\}_{n=1}^{\infty}$ converges to 0 in $L^2(d\mu)$, since

$$\int_\Omega r^2(\tilde{u}_n)\, d\boldsymbol{x} \le 2 \int_\Omega r^2(u_n)\, d\boldsymbol{x} \le \frac{2}{n} \to 0, \quad \text{as } n \to \infty$$

2. The renormalized squared residual distributions

$$d\tilde{\nu}_n \triangleq \frac{r^2(\tilde{u}_n)}{\int_\Omega r^2(\tilde{u}_n(\boldsymbol{x}))\, d\boldsymbol{x}}\, d\mu$$

converges to the uniform distribution $\mu$ in the Wasserstein distance $d_{W^M}$.

$\square$

## A.3 REPLACEMENT OF THE BOUNDEDNESS CONDITION IN THEOREM 4

For the boundedness constraint for "test function" $p$ in 4, we prove that it can be removed in our circumstance. And with the following lemma and its following remark, and Theorem 4, we can obtain our main Theorem 1, which is stated again with its assumption in the following.

**Assumption.** The operator $r$ in equation 7 is a surjection from a function space $E_1(\mathbb{R}^D)$ to $C_c^\infty(\Omega)$, the class of $C^\infty$ functions that are compactly supported on $\Omega$.

**Theorem.** *Let $\mu$ be the Lebesgue measure on $\mathbb{R}^D$, which represents the uniform probability distribution on $\Omega$. In addition, we assume Assumption A1 holds. Then the optimal value of the min-max problem equation 7 is $0$. Moreover, there is a sequence $\{u_n\}_{n=1}^{\infty}$ of functions with $r(u_n) \ne 0$ for all $n$, such that it is an optimization sequence of equation 7, namely,*

$$\lim_{n\to\infty} \mathcal{J}(u_n, p_n) = 0,$$

*for some sequence of functions $\{p_n\}_{n=1}^{\infty}$ satisfying the constraints in equation 7. Meanwhile, this optimization sequence has the following two properties:*

1. *The residual sequence $\{r(u_n)\}_{n=1}^{\infty}$ of $\{u_n\}_{n=1}^{\infty}$ converges to 0 in $L^2(d\mu)$.*

2. *The renormalized squared residual distributions*

$$d\nu_n \triangleq \frac{r^2(u_n)}{\int_\Omega r^2(u_n(\boldsymbol{x}))\, d\boldsymbol{x}}\, d\mu(\boldsymbol{x})$$

*converge to the uniform distribution $\mu$ in the Wasserstein distance $d_{W^M}$.*

Although the residue $r^2$ is renormalized to a probability distribution for the analysis of the algorithm, itself is not a probability distribution, and not treated as so. Actually, in the implementation of our algorithm, the "test function" $p$ is seen as sampling distribution density and the residue $r^2$ is just the PDE operator (or any kind of objective function whose minimum is 0). In the implementation, we establish $p$ as a generative model, that is, an invertible transform between an unknown distribution (an adversarial distribution to the residual distribution if we think the algorithm as a similarity to GANs) and an "easy-to-sample" distribution such as normal or uniform distribution. So we assume $p$ to be the density function of a probability distribution. Under this assumption, we have the following result.

**Lemma 5.** *Let $\Omega$ be a compact subset of $\mathbb{R}^D$. If a positive function $f : \Omega \to \mathbb{R}$ is $K$-Lipschitz continuous, and $f$ is the density function of a probability distribution, namely, $\int_\Omega f\, d\boldsymbol{x} = 1$, then there is some constant $M = M(\Omega, K)$, so that $f \le M$. In other words,*

$$f \le M, \quad \forall f \in \mathcal{S} = \Big\{ f \ge 0 \big| \|f\|_{Lip} \le K, \text{ and } \int_\Omega f\, d\boldsymbol{x} = 1 \Big\}.$$

*Proof.* For any $x, y \in \Omega$, we have

$$0 \le f(\boldsymbol{x}) = f(\boldsymbol{x}) - f(\boldsymbol{y}) + f(\boldsymbol{y}) \le K|\boldsymbol{x} - \boldsymbol{y}| + f(\boldsymbol{y}) \le K\mathcal{D}(\Omega) + f(\boldsymbol{y}),$$

where $\mathcal{D}(\Omega)$ is the diameter of $\Omega$. Taking integral with respect to $\boldsymbol{y}$ over $\Omega$ on both sides, we have

$$0 \le f(\boldsymbol{x})\mu(\Omega) \le K\mathcal{D}(\Omega)\mu(\Omega) + 1,$$

where $\mu(\Omega)$ is the Lebesgue measure (volume) of $\Omega$, that is,

$$0 \le f(\boldsymbol{x}) \le K\mathcal{D}(\Omega) + \frac{1}{\mu(\Omega)}.$$

So we have

$$M = M(\Omega, K) = K\mathcal{D}(\Omega) + \frac{1}{\mu(\Omega)}.$$

$\square$

e The converse of this lemma is also true in the sense that if $f$ is bounded by some constant $M$, then the integral $\int_\Omega f\, d\boldsymbol{x} \le M\mu(\Omega)$, and $f$ can be renormalized into a probability density function with constant $M\mu(\Omega)$. And similar to boundedness for the gradient (or Lipschitz constant) discussed in section 3.3, a constant renormalizer will not affect the training procedure.

## A.4 Deviation of equation 9 and its solution

For a given $r(\boldsymbol{x}; \boldsymbol{\theta})$, consider the following minimization problem:

$$\min_{p_\alpha > 0} \mathcal{L}(p_\alpha) = \beta \int_\Omega |\nabla_{\boldsymbol{x}} p_\alpha|^2 dx - \int_\Omega r^2(\boldsymbol{x};\boldsymbol{\theta})p_\alpha(\boldsymbol{x})dx + \lambda\left(\int_\Omega p_\alpha(dx) - 1\right),$$

where the positivity of $p_\alpha$ is guaranteed by the KRnet and $\lambda$ is the Lagrange multiplier for the mass conservation of PDF. Assuming that $\frac{\partial p_\alpha}{\partial \boldsymbol{n}} = 0$ on the boundary $\partial\Omega$, where $\boldsymbol{n}$ is a unit normal vector on $\partial\Omega$ pointing outward. We have the first-order variation of $\mathcal{L}(p_\alpha)$ for a perturbation function $\delta p(\boldsymbol{x})$

$$\begin{aligned}
\delta\mathcal{L} =& 2\beta \int_\Omega \nabla p_\alpha \cdot \nabla \delta p d\boldsymbol{x} - \int_\Omega r^2 \delta p d\boldsymbol{x} + \lambda \int_\Omega \delta p d\boldsymbol{x} \\
=& 2\beta \left(\int_{\partial\Omega} \delta p \nabla p_\alpha \cdot \boldsymbol{n} d\Gamma - \int_\Omega \delta p \nabla^2 p_\alpha d\boldsymbol{x}\right) - \int_\Omega r^2 \delta p d\boldsymbol{x} + \lambda \int_\Omega \delta p(\boldsymbol{x}) d\boldsymbol{x} \\
=& -2\beta \int_\Omega \delta p \nabla^2 p_\alpha d\boldsymbol{x} - \int_\Omega r^2 \delta p d\boldsymbol{x} + \lambda \int_\Omega \delta p(\boldsymbol{x}) d\boldsymbol{x} \\
=& -\int_\Omega (2\beta\nabla^2 p_\alpha + r^2 - \lambda)\delta p d\boldsymbol{x},
\end{aligned}$$

where we applied integration by parts and the homogeneous Neuman boundary conditions. The optimality condition $\frac{\delta\mathcal{L}}{\delta p} = 0$ yields

$$\begin{cases} 2\beta\nabla^2 p_\alpha(\boldsymbol{x}) + r^2(\boldsymbol{x};\boldsymbol{\theta}) - \lambda = 0, & \boldsymbol{x} \in \Omega, \\ \frac{\partial p_\alpha}{\partial \boldsymbol{n}} = 0, & \boldsymbol{x} \in \partial\Omega. \end{cases} \tag{19}$$

From the compatibility condition for Neumann problems, we have

$$\int_\Omega (r^2(\boldsymbol{x};\boldsymbol{\theta}) - \lambda)d\boldsymbol{x} = 0, \tag{20}$$

which yields that

$$\lambda = \frac{1}{|\Omega|} \int_\Omega r^2(\boldsymbol{x};\boldsymbol{\theta})d\boldsymbol{x}.$$

Assume that $\Omega$ is a bounded domain with smooth boundary. It can be shown that if $r \in H^k(\Omega)$ and $\partial\Omega \in C^{k+2}$ with $k \in \mathbb{N}$, the solution of equation 9 satisfies (Taylor, 2011)

$$\|p_\alpha\|_{H^{k+2}(\Omega)} \le C\|f\|_{H^k(\Omega)},$$

where $f(\boldsymbol{x}) = (\lambda - r^2)/(2\beta)$ and $C > 0$ is a general constant that does not depend on $r$. According to the Sobolev Imbedding Theorem (Adams & John Fournier, 2003),

$$W^{k,1}(\Omega) \to C^{0,1}(\overline{\Omega}),$$

when $D = k - 1$. Thus up to a set of measure zero, we have

$$\|p_{\boldsymbol{\alpha}}\|_{C^{0,1}(\overline{\Omega})} \le C_1 \|p_{\boldsymbol{\alpha}}\|_{W^{k,1}(\Omega)} \le C_2 \|p_{\boldsymbol{\alpha}}\|_{H^k(\Omega)},$$

where $C_1$ and $C_2$ are general constants independent of $p_{\boldsymbol{\alpha}}$. So $p_{\boldsymbol{\alpha}}$ is Lipschitz continuous when the boundary and $r(\boldsymbol{x})$ are sufficiently smooth. However, this also means that the $H_1$ regularization used in equation 8 induces a weaker constraint than the Lipschitz condition in Lemma 5.

### A.5 SUPPLEMENTARY EXPERIMENTS

**About the setting of** $s(\boldsymbol{x})$ **and** $g(\boldsymbol{x})$. The source term $s(\boldsymbol{x})$ is derived by the exact solution, i.e., we can set the source function by plugging the exact solution into the equation to get $s(\boldsymbol{x})$. We set $g(\boldsymbol{x}) = u(\boldsymbol{x})$ since the Dirichlet boundary condition is imposed on $\partial\Omega$.

**Parametric Burgers' Equation**. We also test the proposed AAS method using parametric PDEs that are commonly used in the design of engineering systems and uncertainty quantification. Specifically, we consider the following parametric Burgers' equation, which is a benchmark problem studied in DeepXDE.

$$\frac{\partial u}{\partial t} + u\frac{\partial u}{\partial x} + v\frac{\partial u}{\partial y} = \nu[(\frac{\partial u}{\partial x})^2 + (\frac{\partial u}{\partial y})^2]$$

$$\frac{\partial v}{\partial t} + u\frac{\partial v}{\partial x} + v\frac{\partial v}{\partial y} = \nu[(\frac{\partial v}{\partial x})^2 + (\frac{\partial v}{\partial y})^2]$$

$$x, y \in [0,1], \text{ and } t \in [0,1]$$

where $u$ and $v$ are the velocities along $x$ and $y$ directions respectively, and $\nu \in (0,1]$ is a parameter that represents the kinematic viscosity of fluid. Here, the Dirichlet boundary conditions are imposed on all boundaries. The exact solution is obtained as follows.

$$u(x,y,t) = \frac{3}{4} - \frac{1}{4[1 + \exp((-4x + 4y - t)/(32\nu))]},$$

$$v(x,y,t) = \frac{3}{4} + \frac{1}{4[1 + \exp((-4x + 4y - t)/(32\nu))]},$$

The problem setup space is $\boldsymbol{x} = [t, x, y, \nu]$, i.e., $D = 4$. When $\nu$ is small, solving this problem is quite challenging. We use the proposed AAS method to train a neural network $u_{\boldsymbol{\theta}}(\boldsymbol{x})$ to approximate the solution over the entire space $\boldsymbol{x} = [t, x, y, \nu] \in [0,1]^4$. Figure 6 shows the numerical results, which demonstrate that the proposed AAS method is able to accurately solve this parametric Burgers' equation. We can train the models using the strategy as discussed in Remark 2, i.e., we gradually add the data points to the current training set. AAS with fixed $\beta = 5$ means that we use a similar training strategy as DAS-G presented in (Tang et al., 2023) with a fixed $\beta$, while AAS with decay $\beta = 5$ means that $\beta$ has a decay scheme at every 100 stages with decay rate 0.9. Adding the data points gradually to the current set of random samples is more stable than that of replacing all data points.

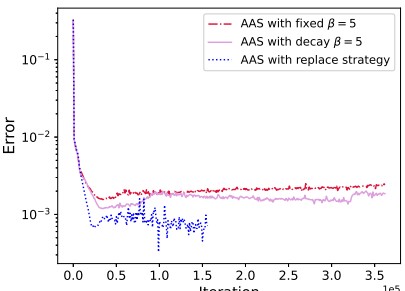 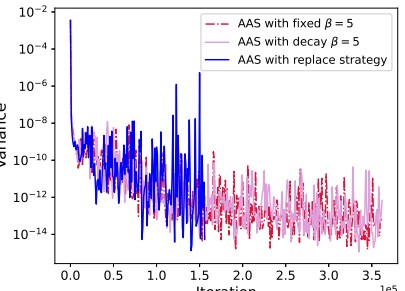

Figure 6: The results of the parametric Burgers' equation. Left: The error behavior. Right: The evolution of the variance.

