# OpenReview forum: "Adversarial Adaptive Sampling: Unify PINN and Optimal Transport for the Approximation of PDEs"
_ICLR.cc/2024/Conference — ICLR 2024 poster_

### Official Review · Reviewer_CyTK · 2023-10-31

**Soundness:** 3 good
**Presentation:** 4 excellent
**Contribution:** 4 excellent
**Rating:** 8
**Confidence:** 4

**Summary:**

In this article, authors raise a significant problem that linked with numerical difficulty is that random samples in the training dataset introduce statistical errors into the functional loss which may become the dominant error in the approximation, and therefore overshadow the modeling capability of the neural network. A new approach was proposed to optimize both the approximate solution and random samples in the training set by using a min max formulation. This approach is called adversarial adaptive sampling (AAS). The main idea of AAS is minimizing the residuals and meanwhile push the residual-induced distribution to a uniform one.  AAS can be divided into two parts. In the maximization part, the deep generative model helps define the difference between the residual-induced distribution and a uniform one using Wasserstein distance. In the minimization one, this difference is minimized together with the residuals.   Also, they used some benchmark test problems of comparison AAS algorithm with another state-of-the-art algorithms as DAS and RAR.

**Strengths:**

The motivation is clear, and the method is novel and interesting. Authors clearly described the theory of proposed method and the algorithm.

**Weaknesses:**

A numerical result is given for one type of PDEs. It would be interesting to see how AAS algorithm performs for other types of PDEs. There is no discussion of further development or limitations of the proposed method.

**Questions:**

What other types of PDEs can the proposed algorithm be applied to?
What are restrictions of the proposed method?

---

> ### Author Response · Authors · 2023-11-17
> **Response to Reviewer CyTK**
>
> 1. A numerical result is given for one type of PDEs. It would be interesting to see how AAS algorithm performs for other types of PDEs. There is no discussion of further development or limitations of the proposed method.
>
> We agree with this reviewer that the performance of AAS needs to be further investigated. The next step is to make AAS more robust by improving the training strategy, and more problems will be considered such as singularly perturbed equations, parametric equations, etc.  We have added the numerical results of parametric Burgers' equation, see Appendix A.5.
>
> 2. What other types of PDEs can the proposed algorithm be applied to? What are restrictions of the proposed method?
>
> AAS is formulated in the framework of the least-squares method. For a certain type of PDE, if the least-squares method is effective, AAS should help to reduce the statistical error. The main restriction comes from the regularization term since it is quite difficult to control the constraint on the Lipschitz norm. More attention needs to be paid to this issue.

---

### Official Review · Reviewer_3BKL · 2023-10-31

**Soundness:** 4 excellent
**Presentation:** 4 excellent
**Contribution:** 4 excellent
**Rating:** 10
**Confidence:** 5

**Summary:**

A uniform approximation error is crucial in representing functions in a given vector base.  The same applies to compound function representations like neural networks. It also crucial where the errors are measured. The Authors combine both aspects, a  Wasserstein distance between a uniform and error distributions,  and an adversarial adaptive sampling of points for the PINN collocation.

The loss is divided to an analytical part that reflects the difference of the neural network from the solution and a statistical part that comes from using a finite sample.  By tuning the residual to become uniform, the density of samples becomes optimally non-uniform.

The method is used for three different PDEs with encouraging results compared to other adaptive learning techniques.

**Strengths:**

An excellent way to bring confidence to the inference with good proofs of convergence for the adaptive learning distributions.

Presentation is clear and not hard to follow.

**Weaknesses:**

Looking at the point clouds one visually can guess that the Delaunay triangulation link length distribution of  the point cloud contains very short distances i.e the point cloud is not locally smooth. Usually building meshes for FEM one prefers same size of elements locally and smaller elements in those areas do not provide more accuracy.

I suppose this is the same for collocation points in PINNs. Currently the point clouds do not look like node distributions for FEM calculus. This is increasingly so, when creating points in the higher dimensions. The amount of wasted calculations increases and may drive this to a curse of high dimensionality.

**Questions:**

Any methods to solve the above problem? A possible remedy could introduce quasi random point distributions where the points do follow the error distribution, but are smoothly enough distributed in oder to avoid computation of the the error on points that are close to each other already without an expectation of significant change in the error distribution.

Could the problem above be solved with couple of iteration steps of repulsive force in the point cloud?

---

> ### Author Response · Authors · 2023-11-17
> **Response to Reviewer 3BKL**
>
> 1. Looking at the point clouds one visually can guess that the Delaunay triangulation link length distribution of the point cloud contains very short distances i.e the point cloud is not locally smooth. Usually building meshes for FEM one prefers same size of elements locally and smaller elements in those areas do not provide more accuracy.
> I suppose this is the same for collocation points in PINNs. Currently the point clouds do not look like node distributions for FEM calculus. This is increasingly so, when creating points in the higher dimensions. The amount of wasted calculations increases and may drive this to a curse of high dimensionality.
>
> Currently the collocation points rely on the best possible PDF that helps smooth the residual profile such that the adaptive sampling can be understood from the viewpoint of variance reduction. We do not have any direct  specific control over the skewness if we relate the points to a mesh. The collocation points are random rather than deterministic. The idea of Delaunay triangulation makes more sense in low-dimensional spaces since points become sparse in high-dimensional spaces and the geometry is also fundamentally different.
>
> 2. Any methods to solve the above problem? A possible remedy could introduce quasi random point distributions where the points do follow the error distribution, but are smoothly enough distributed in oder to avoid computation of the the error on points that are close to each other already without an expectation of significant change in the error distribution.
> Could the problem above be solved with couple of iteration steps of repulsive force in the point cloud?
>
> We agree that quasi-random points can be better if we are able to generalize the intuition of Delaunay triangular to high-dimensional spaces. The idea of repulsive force sounds very interesting, especially for low-dimensional spaces. We need to figure out how the repulsive force affects the variance reduction. Furthermore, the repulsive force will introduce a correlation between random samples, which makes the sample generation more challenging.

---

> > ### Comment · Reviewer_3BKL · 2023-11-22
> >
> > Thank for your clarifications. My remains  that this is an excellent paper. Having similar ideas around is understandable, but the point is that one has found a procedure that is clear and works in practice.
> >
> > On the skewness of the "effective" mesh in PINNS, I would argue that the problem in FEM comes from discretisation of the  derivatives and does not have a counterpart in PINNS. In PINNS the points that are "too" close provide less information (i.e. in the neighbourhood where the Jacobian describes the behaviour of the solution as this information is already provided by a single point).  Then increasing the weight of the loss term would have the same effect than adding points nearby.
> >
> > BTW. Similar considerations could be used to prune out the points too close from a generated i.i.d  random sample.

---

> > > ### Author Response · Authors · 2023-11-22
> > >
> > > Thanks for your appreciation and valuable feedback.

---

### Official Review · Reviewer_ZtMG · 2023-10-31

**Soundness:** 2 fair
**Presentation:** 2 fair
**Contribution:** 3 good
**Rating:** 6
**Confidence:** 3

**Summary:**

In this paper the authors introduce an adaptive sampling technique that for sampling the input points (also referred to as collocation points) for training a PINN architecture on a domain. The main motivation behind it is that for highly irregular PDEs, uniformly sampling collocation points will not work well and result in high residual error for PINNs.

The idea is similar to the adaptive sampling techniques that have been introduced in previous work by Tang et al (2023) and Gao et al (2023), however instead of directly minimizing a pushforward or adaptive sampling, the authors in this paper enforce a wasserstein loss to ensure that the residual error is uniformly distributed across the grid.

That is, the training is done such that the error induced by PINNs is uniformly distributed across the grid, and hence the points are sampled accordingly. The authors operationalize this using a WGAN setup, wherein they train a pushforward map to learn p(x), such the wasserstein-1 loss between the the density of the residual and uniform distribution. This is similar in vein of the adaptive sampling techniques introduced in Gao et al (2023) but the weights are instead learned using a GAN type architecture.

**Strengths:**

The idea to use a GAN type loss with to ensure that the density of the residual is uniform across the domain is very interesting.

In their experiments, the authors are able to do well on very sparse data, i.e., PDEs that have two-peaks, something that PINNs sometimes don’t do well in, and are able to get better results than the previous baselines by Tang et al (2023), thus showing the benefits of using the wasserstein formulation for learning $p_\alpha$.

**Weaknesses:**

In general the work is very similar to that by Tang et al (2023), there the authors are using a network defined using a normalizing flows type of an architecture whereas here the authors are using a GAN instead.

The paper is very hard to read with sometimes the notation and the terms used by the authors for different quantities is not clear. For example

- The authors mention that this methodology achieves variance reduction, what is the proof for that? is it similar in flavor of Tang, et al (2023)
- In Assumption 1, the authors refer to $r$ as an operator. Is that an operator, or the residual between the loss. If the authors mean that $r$ is an operator, then from which function space to what other function space? (I presume it is U x F → U).
- In theorem 1, $\nu$ is used to define the density of the residuals, whereas in the derivation under equation 6, it is defined by $\mu_r$. I think that they are the same quantity, however if not, what is $\nu$ and what is $\mu_r$?
- It is unclear as to what is being approximated by a KRNet?, since the authors first use $p_\alpha$ to define the density that they are trying to train, however, there is no mention of it after section 3.2
- Also, KRNet is not cited in the last paragraph of page 3.
- In equation 9, what is p*?

Few other questions that I think would help with the understanding of the usability of the techniques are:

- What are the implementation details (in terms of number of parameters) etc of the networks approximating the solution $u_\theta$ and approximating the density network, i.,e $p_\alpha$.
- Since the authors are doing a min-max optimization, that is usually hard to train and get right, esp for for PDEs that may have some advective terms.

While the methodology provided by the authors seems to do better than the baselines, given the presentation and the lack of clarity of the precise steps, I think the paper would benefit from a revision

**Questions:**

I have asked most of the questions in the previous sections.

---

> ### Author Response · Authors · 2023-11-17
> **Response to Reviewer ZtMG**
>
> 1. In general the work is very similar to that by Tang et al (2023), there the authors are using a network defined using a normalizing flows type of an architecture whereas here the authors are using a GAN instead.
>
> First, we do not use a GAN in this work. The formulation of AAS is a min-max problem, but not every min-max problem is GAN. This work is similar to the work DAS-PINNs in Tang et al (2023) only in the sense that both consider to use adaptive sampled collocation points to construct a residual-based PDE solver. The adaptive sampling framework of DAS consists of two parts: solving PDEs and refining the collocation points. The two parts in DAS-PINNs are trained separately and not integrated into one objective function, which shares more similarities to adaptive finite element methods.
>
> In this work, we directly put the sampling distribution into the loss functional. Adversarially, if this sampling distribution is the residual distribution, the residual itself will be uniformly distributed, which means the error is evenly spread and thus achieves a variance reduction with respect to the uniform distribution. One big advantage of this work versus DAS is that the training of PDE (residual) and sampling density are unified into one minimax procedure. In the training procedure, we don't have to sample from residual to train the KRnet approximation of sampling density, which could generate additional computational error particularly for high-dimensional problems.
>
> In addition, we also explain the work Gao et al (2023) mentioned by the reviewer. That work uses the same loss and strategy as in Tang et al (2023). The only difference is they use a MCMC technique, instead of a normalizing flow model of KRnet.
>
> 2. The authors mention that this methodology achieves variance reduction, what is the proof for that? is it similar in flavor of Tang, et al (2023)
>
> In AAS, in the training procedure, the residual will approach the uniform distribution, with the balance from the growth of the adversarial sampling density function $p$. This will guarantee a variance reduction of residual, because if the residual is uniformly distributed, it is constant in the domain, and a constant function has $0$ variance with respect to any probability measure. This is why we didn't explain it in the paper.
>
> 3. In Assumption 1, the authors refer to $r$ as an operator. Is that an operator, or the residual between the loss. If the authors mean that $r$ is an operator, then from which function space to what other function space? (I presume it is $U \times F \rightarrow U$).
>
> Note that $r$ is the residual of PDEs. Given the partial differential operator $\mathcal{L}$, and the right-hand term $s$, for any function $u \in C^{\infty}(\Omega)$, then the residual defines an operator, which is interpreted as
>     $r(u) = \mathcal{L}u - s$.
>
> 4. In theorem 1, $\mu$ is used to define the density of the residuals, whereas in the derivation under equation 6, it is defined by $\mu_r$. I think that they are the same quantity, however if not, what is $\mu$ and what is $\mu_r$?
>
> In theorem 1, $\mu$ represents the uniform distribution. $\nu$ represents the residual-induced distribution. To avoid confusion, we have revised the manuscript, where $\nu$ is replaced by $\mu_r$.
>
> 5. It is unclear as to what is being approximated by a KRNet?, since the authors first use $p_\alpha$ to define the density that they are trying to train, however, there is no mention of it after section 3.2
>
> Here, $p_{\boldsymbol{\alpha}}$ is a PDF modeled by KRnet. Solving PDEs with neural networks needs to generate collocation points in the computational domain for training. These collocation points are drawn from $p_{\boldsymbol{\alpha}}$ defined by KRnet—different sets of collocation points correspond to different KRnets. So, the maximization step in equation 8 is the training procedure for the KRnet modeled pdf, see section 3.3.
>
> 6. Also, KRnet is not cited in the last paragraph of page 3.
>
> We here use a bounded KRnet, and the citation is on page 4: ``The bounded KRnet can be achieved by adding a logistic transformation layer (Tang et al., 2023) or a new coupling layer proposed in (Zeng et al., 2023)". We have also added a reference on page 3 in the revised manuscript.
>
> 7. In equation 9, what is $p^*$?
>
> For a fixed $u_{\boldsymbol{\theta}}$, equation (9), the Euler–Lagrange equation for the functional in equation (8), gives the necessary condition for the existence of optimal $p*$ of the maximization step in equation (8), which is a standard technique called the calculus of variations in analysis. We write this equation to illustrate that the maximization step is well-posed after adding a gradient penalty term (see equation 8), which is a weakened condition for the Lipschitz norm condition. The regularization of it is then guaranteed, for otherwise, the maximization step will yield a delta measure (see section 3.1).

---

> > ### Author Response · Authors · 2023-11-17
> > **Response to Reviewer ZtMG**
> >
> > 8. What are the implementation details (in terms of number of parameters) etc of the networks approximating the solution $u_\theta$ and approximating the density network, i.,e $p_\alpha$.
> >
> > Due to the page limitation, we do not present the details of the network structure. All the models are set to be the same as those in DAS-PINNs. We have stated this issue in the revised manuscript.
> >
> > 9. Since the authors are doing a min-max optimization, that is usually hard to train and get right, esp for for PDEs that may have some advective terms.
> >
> > In Appendix A.5, we have added the numerical results of a parametric Burgers' problem. By the way, we do not claim that the AAS method can beat classical methods. It is just an improvement of residual-based machine learning solvers, and by introducing adaptivity, it is expected to solve problems (mainly PDEs) with sigularities more efficiently and accurately in high dimensional problems. We use the benchmark problems to illustrate one thing: one should pay attention to the sampling method when training PINN, especially for low-regularity or high-dimensional problems, just like adaptive mesh and moving mesh were also introduced to classical methods such as finite element methods. The sampling strategy of this work can be coupled with other techniques based on neural networks to solve PDEs. This work proposes a unified framework for adaptive sampling, with the solution and the sampling networks being trained simultaneously. However, we do not claim that adaptive sampling can resolve every issue.

---

> > > ### Comment · Reviewer_ZtMG · 2023-11-22
> > > **Thank you for the response**
> > >
> > > The authors have answered a lot of my questions well in their response. I would recommend adding the appropriate clarifications in the final version of the paper. Thanks!

---

> > > > ### Author Response · Authors · 2023-11-23
> > > >
> > > > Thanks for your reply. We will check the clarifications in the revised manuscript and proofread it again.

---

### Official Review · Reviewer_YsSo · 2023-11-01

**Soundness:** 2 fair
**Presentation:** 1 poor
**Contribution:** 2 fair
**Rating:** 5
**Confidence:** 4

**Summary:**

The paper proposes a new objective to sample collocation points for PINNs adaptively. Specifically, the collocation points for training the PINN are sampled from a normalizing flow with soft Lipschitz constraint, enforced by Sobolev regularization. The PINNs and normalizing flows are optimized in an alternating fashion, where the normalizing flow is trained to maximize the expected PINN residual (w.r.t. to the distribution given by the pushforward of the normalizing flow). This formulation is then connected to the dual form of an optimal transport problem. In this context, it is shown that there exists an optimal solution to the proposed min-max problem where the (normalized) squared residual converges to a uniform distribution in the Wasserstein distance. The approach's effectiveness is further demonstrated on two toy problems and one high-dimensional nonlinear PDE and compared to three related methods.

**Strengths:**

1) Tackling adaptive sampling for PINNs using optimal transport seems to be a promising direction.
2) The proposed objective seems to be novel and can bring numerical benefits on the considered examples.

**Weaknesses:**

The theoretical as well as numerical contributions need to be significantly improved:

1) Further challenging problems (as in Subsection 5.3), as well as baselines (as enumerated in the section on related works), are needed to judge the performance of the proposed algorithm.
2) "[...] which is the first time to minimize the residual and seek the optimal training set simultaneously for PINN.": Analogous to DAS-PINNs, it seems that the final algorithm is still alternating between optimizing the two networks, see also Question 2) below. In general, it should be made more explicit what the novelty of the present work is and how it theoretically compares to related work.
3) For the numerical results, training times and standard deviations (w.r.t. different seeds) are missing. Especially given that the adversarial training slows down training.
4) The loss for learning the normalizing flow seems to use the REINFORCE trick, which, however, is known to suffer from high variance.
5) Since the residual continuously changes over the course of optimization, it should be better motivated what the advantage of learning a generative modeling is (as compared to just using a method to sample, e.g., according to squared residual)?
6) It seems that the precise connection to optimal transport remains a bit unclear:
	* "So the minimization step will reduce not only the residual but also the Wasserstein distance between $\mu_r$ and the uniform distribution". Since there is only an upper bound shown on page 6, it is unclear why the minimization step is guaranteed to decrease the Wasserstein distance.
	* "The evolution of the residual-induced distribution has a clear path": According to the theorem, there only *exists* such a path, and it is not clear whether this path is taken by GD. It would be good to have at least plots of the residual evolution for all experiments.

**Minor issues:**
1) use `\citep`.
2) In Eq. (4), min-max should be written on both sides of the equation.
3) What is a 'proper' constraint?
4) The last paragraph in Section 3 seems a bit convoluted. The choice for minimal variance could just be motivated by the optimal choice for importance sampling.
5) It is unclear why we would consider $I^D$ instead of $\Omega$ when the KRnet is first introduced.
6) The text for the figures of the training set is barely readable.
7) typo: resultd

**Questions:**

1) Why is RAR not tested for the PDEs in (12) and (13)?
2) Why can we not use a single backward pass on the loss in (11) and update both $\theta$ and $\alpha$ (using decent and ascent, respectively).

---

> ### Author Response · Authors · 2023-11-17
> **Response to Reviewer YsSo**
>
> 1. Further challenging problems (as in Subsection 5.3), ... are needed to judge the performance of the proposed algorithm.
>
> We have added the additional numerical experiments (parametric Burgers' equations) to Appendix A.5 when we submitted the original manuscript. We do not claim that the AAS method can beat classical methods. We use the benchmark problems to illustrate one thing: one should pay attention to the sampling method when training PINN, especially for low-regularity or high-dimensional problems. However, we do not claim that adaptive sampling can resolve every issue. Adaptive sampling for PINN is a new direction, where DAS-PINNs is a recent work, and we have already used DAS-G and DAS-R as baselines. It is not a computer vision or NLP task, which requires a lot of old baselines.
>
> 2. Analogous to DAS-PINNs, it seems that the final algorithm is still alternating between optimizing the two networks, see also Question 2) below. In general, it should be made more explicit what the novelty of the present work ...
>
> The current algorithm allows the two models (PINN and KRnet) to evolve at the same time. In the work of DAS-PINNs, the style of adaptive sampling is like that of the classical methods such as adaptive finite element methods. That is, the procedure of adaptive sampling in DAS-PINNs is like $\mathsf{solve} \rightarrow \mathsf{refine} \rightarrow \mathsf{solve} \rightarrow \ldots$. Simply speaking, given a training set, we need to train PINN with a large number of epochs to ensure that we have ``solved" the PDE, and then we use the KRnet-induced distribution to approximate the residual-induced distribution by minimizing the KL divergence between them. However, in this work, we do not require the minimization step to have a large number of epochs. Moreover, the min-max formulation in AAS implicitly pushes the residual-induced distribution to a uniform one without any KL divergence-like formulation, while there is an explicit minimization step about KL divergence in DAS-PINNs. In summary, AAS's formulation and training style are different from DAS-PINNs.
>
> 3. For the numerical results, training times and standard deviations (w.r.t. different seeds) are missing. Especially ...
>
> Thanks for the suggestion. Based on our available computational resources, we have taken three runs with different random seeds for the proposed AAS method and computed the mean error and the standard deviation (std) of the three runs. Also, we record the mean training time of the three runs for the three test problems presented in the manuscript. The results are as follows.
> |   | Mean error with std   | Training time |
> |  ----  | ----  | ----  |
> | Equation 12  | 3.1754e-05 $\\pm$ 1.1272e-05 |   4.82 hours |
> | Equation 13  |  6.4194e-04 $\\pm$ 4.6891e-04 |  5.41 hours |
> | Equation 14  |  0.0024 $\\pm$ 0.0015 |  15.74 hours |
>
> 4. The loss for learning the normalizing flow seems to use the REINFORCE trick, ... suffer from high variance.
>
> We are not quite sure about what the reviewer means by ``use the REINFORCE trick". The normalizing flow model generates samples for training PINN. Still, we do not require the flow model to have a high-accuracy performance since our goal is to solve PDEs instead of density estimation. It is enough if the normalizing flow model can capture where the low regularity of the solution is. In addition, other deep generative models with an explicit PDF can be applied here.
>
> 5. Since the residual continuously changes over the course of optimization, it should be better motivated what the advantage of learning a generative modeling is (as compared to just using a method to sample, e.g., according to squared residual)?
>
> Sampling data points according to the residual is suitable for low-dimensional problems. For high-dimensional problems, using sampling methods such as MCMC is not efficient, and this is why deep generative models (such as GAN, VAE, Flow, and diffusion models) are proposed.
>
> 6. Since there is only an upper bound shown on page 6, it is unclear why the minimization step is guaranteed to decrease the Wasserstein distance.
>
> This is just an illustrative analysis to show how the idea works. For more details, see Appendix A.1-A.4. We have provided the detailed analysis about this question.

---

> > ### Author Response · Authors · 2023-11-17
> > **Response to Reviewer YsSo**
> >
> > 7. According to the theorem, there only exists such a path, and it is not clear whether this path is taken by GD. It would be good to have at least plots of the residual evolution for all experiments.
> >
> > The theorem tells us that the residual-induced distribution converges to the uniform distribution. As long as the maximization of the sequence $\{p_n\}$ is efficiently solved, by substituting it into the integral, the minimization process is to force the residual to converge to the uniform distribution. Then, GD is just one way to make the parameterized network function to achieve this process, as in any other machine learning methods. In addition, the variance of the residual function values will converge to zero, i.e., the residual at the collocation points are almost the same. We have plotted the variance of the residual in the manuscript (see Figure 2(b), the left plot of Figure 4, and Figure 5(b)).
> >
> > 8. use citep.
> >
> > Thanks for pointing out this issue. We have fixed it in the revised manuscript.
> >
> > 9. In Eq. (4), min-max should be written on both sides of the equation.
> >
> > Thanks for pointing out this issue. We have fixed it.
> >
> > 10. What is a 'proper' constraint?
> >
> > The proper constraint for the PDF model $p_{\boldsymbol{\alpha}}$ is used to make the min-max formulation well-posed. Otherwise, the maximization step will simply yield a delta measure (see section 3.1). To remedy this issue, we can enforce a constraint on $p_{\\boldsymbol{\\alpha}}$. Here, the constraint is $p_{\\boldsymbol{\\alpha}} \\geq 0, \\int_{\\Omega} p_{\\boldsymbol{\\alpha}} d \\boldsymbol{x} =1, ||p_{\\boldsymbol{\\alpha}}||_{Lip} \\leq 1$, see section 3.2.
> >
> > 11. The last paragraph in Section 3 seems a bit convoluted. The choice for minimal variance could just be motivated by the optimal choice for importance sampling.
> >
> > For a given set of collocation points, the residual function values at these collocation points form a vector. The variance of this vector is what we mean in section 3. We expect to choose an ``optimal" set of collocation points to make the variance small, i.e., the residual at each collocation point is almost the same. This idea is similar to the classical adaptive finite element method, where mesh refinement/coarsening is supposed to make the approximation error nearly uniform.
> >
> > 12. It is unclear why we would consider $I^D$ instead of $\Omega$ when the KRnet is first introduced.
> >
> > Thanks for pointing it out. We focus on the problem that the solution has low-regularity, which is used to demonstrate the important role of adaptive sampling in the framework of PINN. In this work, the computational domain is restricted to $[-1,1]^D$, i.e., $\\Omega = [-1,1]^D$. This is because we want to rule out the other factors such as complex boundaries to restrict our attention to adaptive sampling. We have added the comments (``In this work, we consider $\\Omega = I^D$ for simplicity") to the revised manuscript.
> >
> > 13. The text for the figures of the training set is barely readable.
> >
> > Thanks for pointing it out. We will replot these figures by adjusting its font size to make the text more clear.
> >
> > 14. typo: resultd
> >
> > We have fixed this typo and proofread the manuscript again.
> >
> > 15. Why is RAR not tested for the PDEs in (12) and (13)?
> >
> > We mainly use low- dimensional and low-regularity test problems to demonstrate that the sampling strategy affects significantly the performance of neural network approximation if the residual is strongly localized. For low-dimensional problems, RAR is simple, easy to implement, and sufficient. In RAR, candidate points to add into the current training set are also drawn from the uniform distribution. The candidate points who have large residuals are selected to add into the training set to refine the collocation points. However, for high-dimensional problems, generating the candidate points by the uniform distribution is not efficient, sometimes failed, but AAS and DAS can efficiently generate the data points to refine the collocation points thanks to the deep generative model. This is also discussed by the review paper \emph{A comprehensive study of non-
> > adaptive and residual-based adaptive sampling for physics-informed neural networks. Computer
> > Methods in Applied Mechanics and Engineering, 403:115671, 2023.} See the conclusion of this paper. The authors cite the work of DAS-PINNs: “In this study, we sample residual points in RAD and RAR-D by using a brute-force approach, which is simple, easy to implement, and suﬀicient for many PDEs. However, for high-dimensional problems, we need to use other methods, such as generative adversarial networks (GANs) [46], as was done in Ref. [41]”, where Ref. [41] indicates the DAS method.
> >
> > 16. Why can we not use a single backward pass on the loss in (11) and update both $\theta$ and $\alpha$ (using decent and ascent, respectively)
> >
> > It will lead to divergence. The training style is similar to WGAN, where the two model parameters are updated alternatively to ensure convergence.

---

> > > ### Comment · Reviewer_YsSo · 2023-11-23
> > >
> > > Thank you for your response, the clarifications, and the additional experiments. I raised my score accordingly; however, I believe the empirical validation and presentation still need to be improved. Let me specifically elaborate on the empirical validation:
> > >
> > > *Empirical evidence:* Thank you for providing runtimes for the experiments; as hinted at in my review, it would be good to compare to the runtimes of the baselines. Further experiments could have validated the statements in the rebuttal as well:
> > >
> > > 1. "For high-dimensional problems, using sampling methods such as MCMC is not efficient, and this is why deep generative models (such as GAN, VAE, Flow, and diffusion models) are proposed.": I agree that DL models provide promising alternatives; however, in most applied fields, MCMC methods (and extensions such as SMC, AIS, ...) are still the gold standard, even for high-dimensional problems.
> > >
> > > 2. "However, for high-dimensional problems, generating the candidate points by the uniform distribution is not efficient, sometimes failed, but AAS and DAS can efficiently generate the data points to refine the collocation points thanks to the deep generative model."
> > >
> > > In both cases, the argument evolves around high-dimensional problems, but there seems to be only a single high-dim. (10d) problem in the current set of experiments. Given the claim that these baselines perform worse, including them as (weak) baselines would have been easy. I do not think the statement "It is not a computer vision or NLP task, which requires a lot of old baselines." justifies not presenting further experiments and baselines. Based on the lack of ablation studies or further high-dimensional problems, it is hard to judge the contribution of the new method.

---

> ### Author Response · Authors · 2023-11-23
>
> Thanks for your reply. We agree with this reviewer that MCMC methods (extensions such as SMC, AIS, etc.) are still the gold standard. However, our goal here is to solve PDEs instead of sampling. So, we need to generate samples quickly during the adaptive sampling procedure.

---

### Meta-Review · Area_Chair_s9CG · 2023-12-10

**Metareview:**

The paper proposes the way to learn a sampling measure for training PINNs. The measure for sampling is trained simultaneously, and an additional regularizer is introduced for the measure (Lipschitz continuity). The model is then trained in the adversarial way, by minimizing over PINN and maximizing over the measure. Theorem is proven that the renormalized squared residual convergences to uniform distribution.

Three numerical examples are shown

Strengths:
1) Strict mathematical formulation of optimal sampling
2) Improvement in the convergence rate

Weaknesses:
1) Rather model numerical examples
2) No comparison with classical adaptive sampling (meshing) techniques for PDEs. They are potentially much more efficient for small dimensions (will not work for high dimensions).
3) Squared residual loss is not a suitable loss function for PDEs, for example, for Poisson equation, where the solution lies in another function space, so the approach should be modified for correct loss functions (such as energy function).
4) Only one type of PDE is investigated.

**Justification For Why Not Higher Score:**

Two reviewers put extremely high score (one even 10). If you look at those reviews, they are rather short and non-specific. More negative reviews are more detailed, so the paper is actually closer to borderline in fact, rather than to spotlight/oral.

**Justification For Why Not Lower Score:**

The idea is interesting and has simple but correct theory behind it.

---

### Decision · Program_Chairs · 2024-01-16

Accept (poster)